# 🐍 ConDa: Fast Federated Unlearning with Contribution Dampening

## Abstract

Federated learning (FL) has enabled collaborative model training across decentralized data sources or clients. While adding new participants to a shared model does not pose great technical hurdles, the removal of a participant and their related information contained in the shared model remains a challenge. To address this problem, federated unlearning has emerged as a critical research direction, seeking to remove information from globally trained models without harming the model performance on the remaining data. Most modern federated unlearning methods use costly approaches such as the use of remaining clients data to retrain the global model or methods that would require heavy computation on client or server side. We introduce Contribution Dampening (ConDa), a framework that performs efficient unlearning by tracking down the parameters which affect the global model for each client and performs synaptic dampening on the parameters of the global model that have privacy infringing contributions from the forgetting client. Our technique does not require clients data or any kind of retraining and it does not put any computational overhead on either the client or server side. We perform experiments on multiple datasets and demonstrate that ConDa is effective to forget a client's data. In experiments conducted on the MNIST, CIFAR10, and CIFAR100 datasets, ConDa proves to be the fastest federated unlearning method, outperforming the nearest state-of-the-art approach by at least 100×. Our emphasis is on the non-IID Federated Learning setting, which presents the greatest challenge for unlearning. Additionally, we validate ConDa's robustness through backdoor and membership inference attacks. We envision this work as a crucial component for FL in adhering to legal and ethical requirements.

## 1 Introduction

Federated learning (FL) has enabled the collaborative training of machine learning models across decentralized data sources or clients, facilitating the development of more accurate and robust models. Local clients benefit by getting an aggregated and more powerful model without sharing their private data. However, this collaborative approach also raises concerns about the integrity of the model (Yang et al., 2019) when requested to unlearn data from certain clients. In federated learning, models are trained on data from multiple clients, and the global model may inadvertently memorize information from individual data sources. This poses significant challenges when a client requests to remove their contribution from the global model due to contractual, legal compliance or privacy reasons. The global model may retain information about the client. Federated unlearning (Gao et al., 2022; Liu et al., 2021) seeks to address this challenge by developing methods to remove information from globally trained models. The unlearning methods play a crucial role in supporting the 'right to be forgotten' paradigm as required in various data protection regulations (Voigt & Von dem Bussche, 2017; Harding et al., 2019). Such data removal might also be required when any client's data is outdated (Kurmanji et al., 2023), erroneous (Tanno et al., 2022; Schoepf et al., 2024a), or poisoned (Goel et al., 2024; Schoepf et al., 2024b). However, deleting the client's contribution effectively is a difficult task in existing FL frameworks.

**Motivation of this work.** One of the drawbacks in the existing federated unlearning systems (Gao et al., 2022; Liu et al., 2021) is that the methods require help of remaining clients (that we wish to retain) for further updates and retraining purposes. Most of the existing methods have incorporated

*remaining clients* so that they can unlearn their global model. This approach is inefficient and expensive as the burden on unlearning one client or subset of one client's data should not be transferred to *remaining clients* that would lead to multiple steps of retraining and communication rounds. Moreover, for the retraining or updates, it can not be assumed that *remaining clients* will keep holding the data that they used while training their local model. We can choose to remove the contribution of forgetting client only. But while trying to erase the contribution of the local model from the global model, it may affect the global model parameters that are contributed by other clients as well. Another key factor is the clients' data distribution. In practice, client data is not independent and identically distributed (IID); instead, it is unevenly distributed across classes (Zhao et al., 2018). Some methods have improved retraining efficiency. For example, Liu et al. (2021) reduces the number of retraining rounds, while asynchronous federated unlearning (Su & Li, 2023) divides clients into clusters, limiting retraining to relevant clusters. Wu et al. (2022) avoids retraining from scratch but requires the server to perform knowledge distillation with additional unlabeled data. Federated unlearning with momentum degradation (Zhao et al., 2023) erases the forgetting client's contribution, adjusting the model to approach one retrained on the remaining data.

**Our work.** The existing federated unlearning methods rely on costly approaches, such as retraining the global model using the remaining clients' data (Gao et al., 2022; Su & Li, 2023; Liu et al., 2021) or employing computationally expensive methods on the client or server side Wu et al. (2022). These approaches not only incur significant computational overhead but also may compromise the privacy and security of the remaining clients' data. To overcome these limitations, there is a pressing need for efficient and privacy-preserving federated unlearning methods that can effectively remove client-specific information from the global model without compromising its performance. In this paper, we propose Contribution Dampening (CONDA) that enables efficient federated unlearning by tracking the parameters that affect the global model for each client and performing synaptic dampening on the parameters of the global model that have privacy-infringing contributions from the forgetting client. CONDA unlearns a client's contribution from the global model without the need to retrain with remaining clients data as well as not put significant computation or communication overhead to the remaining clients. Federated unlearning may take place on three levels: class unlearning, client unlearning, and sample unlearning. Our method focuses on client-level unlearning. If similar classes of data (as the forget client) are available with other clients as well, the accuracy intuitively decreases for those clients as well. We demonstrate the effectiveness of CONDA through experiments on multiple datasets, showcasing its ability to efficiently forget a client's data while maintaining the model's performance.

The main contributions are summarized as follows:

- **CONDA Framework for Federated Unlearning:** CONDA enables efficient removal of client-specific information from the global model by tracking and selectively dampening parameters updated by the "forget" client while preserving those updated by retained clients.

- **Data-Free and Efficient Unlearning:** CONDA achieves unlearning without retraining or needing access to the training data from remaining clients, minimizing computational overhead and maintaining client privacy.

- **Experimental Validation:** Our experiments on multiple datasets demonstrate CONDA's ability to effectively remove client data while maintaining model performance, outperforming existing unlearning methods.

## 2 PRELIMINARIES

Federated learning (FL) is a machine learning approach where multiple clients (e.g., organizations, mobile devices, etc.) collaboratively train a model while keeping their data stored locally. A central federated server orchestrates the process by selecting eligible clients for each training round, receiving their locally computed model updates, and aggregating these updates to refine the global model. This process continues iteratively until the model converges. We briefly introduce the *unlearning* problem within a FL framework which we denote as Federated Unlearning (FUL) throughout this paper. We examine a situation where a single or multiple clients request a service provider to remove their data from the model to safeguard user privacy and mitigate legal risks. We define the issue of unlearning the *target clients* in FL in this context.

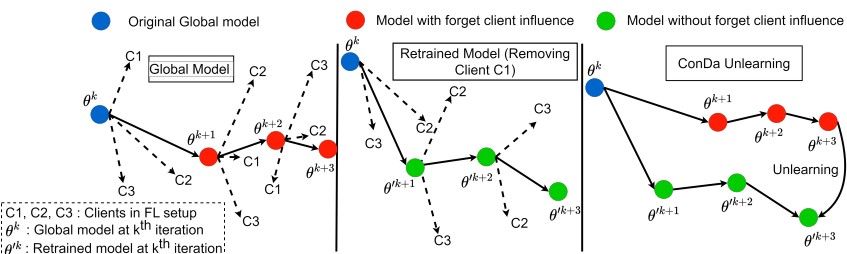

Figure 1: The process of *Client* unlearning in a federated learning (FL) setting is depicted. We also show the efficient nature of the proposed CONDA for federated unlearning.

**Federated Unlearning.** We initialize a global model $\mathcal{M}$ on the central server with parameters $\theta^k$, where $k$ spans the set $E$ representing the global communication rounds. The model is then dispatched to clients for local training, where each client updates the model independently. Following local optimization, the client models are aggregated to form the updated global model. Each iteration, this process is repeated using the new aggregated global model. We visually depict the process of unlearning in FL in Figure 1.

Consider a global model $\mathcal{M}$ trained on data $\mathcal{D}$ distributed across $N$ clients $\mathbf{C} = \{c_1, c_2, \ldots, c_N\}$, where each client $c_n$ contributes local updates $\theta_n$ obtained by minimizing a local loss function $\mathcal{L}(\theta^k; d_n)$ on their respective data $d_n$:

$$\theta_n = \arg\min_\theta \mathcal{L}(\theta^k; d_n), \quad \forall n \in \{1, 2, \ldots, N\}. \tag{1}$$

These local updates are aggregated to update the global model at each communication round $t$ as:

$$\mathcal{M}(\theta)^{k+1} = \frac{1}{N} \sum_{n=1}^{N} \theta_n^k. \tag{2}$$

**Definition 1.** *Let $c_i$ be the forget client who opts out of the FL setup and wants its data removed from the global model $\mathcal{M}$. Then federated unlearning method $\mathcal{FUL}$ aims to update the global model such that it behaves as though the training data $d_i$ from client $c_i$ was never used. This requires adjusting the global model parameters $\theta$ by removing the influence of $\theta_i$ from the aggregation process, represented as:*

$$\mathcal{M}(\theta)_{unlearned}^{k+1} = \frac{1}{N-1} \sum_{\substack{n=1 \\ n \neq i}}^{N} \theta_n^k \tag{3}$$

*where the contributions of client $c_i$ are excluded, effectively reconstructing the model as if client $c_i$'s data had not been incorporated into the training.*

**Challenges.** Following are the crucial challenges in FUL: ① *Machine Unlearning Vs Federated Unlearning:* Machine unlearning methods typically rely on having access to the complete training data. However, this assumption is invalid in FL, where the number of participating clients/devices is often significantly smaller than the total available clients/devices. ② *IID Vs non-IID training data:* The IID data ensures that each client has data that represents the overall population, making it easier for the global model to aggregate local updates effectively. Training is faster, and the global model converges with fewer conflicts. In contrast, non-IID data occurs when clients have vastly different or skewed data distributions, which can lead to biased local models. These biases make it difficult for the global model to generalize well across all clients, resulting in slower training, inconsistent updates, and lower overall performance. *In this paper, we work with non-IID FL setup which is extremely challenging for unlearning.*

## 3 PROPOSED METHOD

We present CONDA, a rapid federated unlearning technique utilizing contribution dampening, which eliminates the need for fine-tuning the global model and significantly reduces computational overhead.

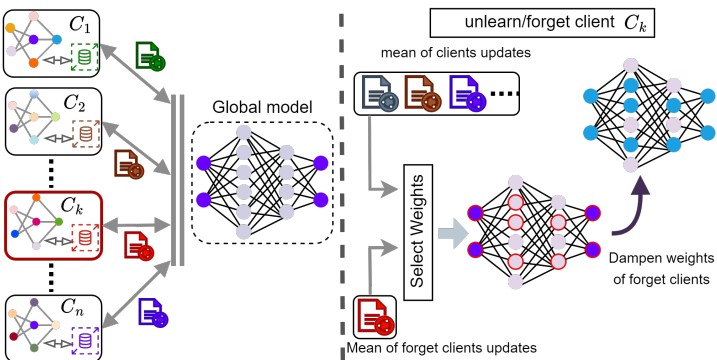

Figure 2: The proposed Contributed Dampening (CONDA) method for federated unlearning.

**Intuition.** This work builds upon the concept of Selective Synaptic Dampening used in traditional machine unlearning (Foster et al., 2024; Feldman, 2020; Stephenson et al., 2021) to create a lightweight machine unlearning method that overcomes the additional challenges introduced by the decentralized federated unlearning setting. The existing work leverages the intuition that specific model parameters are crucial for memorizing certain training examples (forget set, $D_f$) but are not as significant for the remaining data (retain set, $D_r$) (Feldman, 2020). These specialized parameters, which essentially memorize specific data that falls outside of model generalization, are essential for forgetting targeted data without compromising model performance on the broader dataset. In federated unlearning, however, the challenge is compounded by the decentralized nature of data, with information distributed across multiple clients. Our approach adapts this principle to selectively dampen parameters influenced by individual clients while preserving the generalization ability of the global model. This allows for efficient unlearning in federated environments, ensuring compliance while maintaining model performance across diverse client data.

CONDA identifies the global model parameters most impacted by the forget client and dampens them to achieve effective unlearning while preserving the parameters essential for maintaining the model's performance on the retained data. The unlearning process operates entirely on the server side, placing no computational or data-related burden on the clients. The goal is to remove the influence of the forget client without compromising the accuracy of the updated global model. The complete framework is depicted in Figure 2.

We collect each client's contribution to the global model in the form of gradient updates from every communication round in which they participate. Let $\nabla\theta$ represent the average of the gradient updates (contribution) from each client over $E$ commnuication rounds.

$$\nabla\theta_n = \frac{1}{E}\sum_{k=1}^{E}\nabla\theta_n^k \qquad \forall n \in N \tag{4}$$

where $\nabla\theta_n^k = \theta_n^k - \theta^k$ is the gradient update made by client n at communication round k.

When a *client* requests to revoke their data and discontinue participation in FL, we leverage the stored gradient updates collected during the learning process to efficiently unlearn the client's contributions. To differentiate between contributions from all clients and those specifically from the forget clients, we define two sets: Let $C$ denote the set of all clients and $C_f$ denote the subset of clients requesting their data/contribution removal from the global model i.e., set of forget clients. The average contribution of all clients, $\Phi_C$, is computed as:

$$\Phi_C = \frac{1}{|C|}\sum_{c \in C}\nabla\theta_c \tag{5}$$

Similarly, the average contribution of forget clients, $\Phi_{C_f}$, is computed as:

**Algorithm 1** CONDA Federated Unlearning

1: $\theta$: global model parameters
2: $\nabla\theta_n$: average gradient updates for each client
3: $C$: set of clients
4: $C_f$: set of forget clients
5: $\lambda$: dampening constant
6: $\alpha$: cut-off for dampening
7: $U$: dampening upper bound
8: $\Phi_C = \frac{1}{|C|}\sum_{n \in C}\nabla\theta_n$ (average for all clients)
9: $\Phi_{C_f} = \frac{1}{|C_f|}\sum_{n \in C_f}\nabla\theta_n$ (average for forget clients)
10: **Compute** ratio $= \frac{\Phi_C}{\Phi_{C_f}}$
11: **Compute** $\zeta = \lambda \cdot$ ratio
12: **Set** $\beta = \min(\zeta, U)$
13: **for** each $i \in |\theta|$ **do**
14:     **if** $\beta_i < (\alpha \cdot \text{ratio}_i)$ **then**
15:         $\theta'_i = \beta_i \cdot \theta_i$
16:     **end if**
17: **end for**
18: **return** $\theta'$

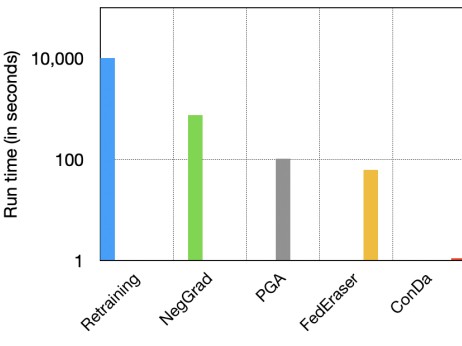

Figure 3: We show the runtime comparison of the proposed CONDA and with the re-trained model, negative gradient, PGA, and FedEraser on ResNet18+CIFAR-10. The Y-axis is on log scale and time is reported in seconds

$$\Phi_{Cf} = \frac{1}{|C_f|}\sum_{c \in C_f}\nabla\theta_c \tag{6}$$

To facilitate the unlearning process, we introduce a dampening factor, denoted as $\zeta$, which is calculated based on the contributions of both the forget clients and all clients. An important consideration in FL is that clients may possess overlapping data, meaning their contributions to the model are not entirely independent. This overlap complicates the unlearning process, as simply removing the net influence of forget clients may leave residual effects from similar data held by retained clients. It is crucial to retain the beneficial contributions from these clients to preserve model accuracy. To address this, we compute the gross influence of the forget clients, isolating their total contribution to the global model while ensuring that the retained clients' influence is preserved.

$$\zeta = \lambda * \frac{\Phi_C}{\Phi_{Cf}} \tag{7}$$

Here, $\lambda$ is dampening constant which is hyperparameter to control amount of forgetting in the global model.

Though we utilize a dampening factor to unlearn the contributions of forget clients, dampening the entire models parameters would lead to catastrophic forgetting and render the resulting model useless. To address this, we introduce new selection criterion in our dampening factor to control which parameters should be dampened for efficient unlearning while preserving the contributions of retain clients. We introduce a cut-off ratio $\alpha$ to control the extent of modification on the global model. This hyperparameter serves as a regularization term in our method to ensure that only parameters that are disproportionally influenced by the forget client are dampened. It represents a boundary between significant and insignificant contributions from forget clients.

$$\zeta = \begin{cases} \lambda * \frac{\Phi_C}{\Phi_{Cf}} & \text{if}\frac{\Phi_C}{\Phi_{Cf}} < \alpha \\ 0 & \text{if}\frac{\Phi_C}{\Phi_{Cf}} \geq \alpha \end{cases} \tag{8}$$

Table 1: Distribution of data samples (CIFAR-10) from each class across 10 different clients in a federated learning setup.

| Class | 0 | 1 | 2 | 3 | 4 | 5 | 6 | 7 | 8 | 9 |
|-------|---|---|---|---|---|---|---|---|---|---|
| *Client 0* | 385 | 13 | 117 | 397 | 380 | 405 | 61 | 905 | 1213 | 1803 |
| *Client 1* | 79 | 72 | 424 | 17 | 2615 | 2 | 986 | 208 | 1401 | 0 |
| *Client 2* | 1193 | 60 | 260 | 16 | 990 | 25 | 522 | 22 | 337 | 860 |
| *Client 3* | 1808 | 72 | 29 | 200 | 2 | 47 | 395 | 54 | 11 | 670 |
| *Client 4* | 143 | 1798 | 1474 | 62 | 38 | 871 | 147 | 15 | 35 | 312 |
| *Client 5* | 120 | 147 | 908 | 2157 | 106 | 235 | 53 | 45 | 1411 | 0 |
| *Client 6* | 126 | 9 | 117 | 236 | 163 | 1671 | 1263 | 2591 | 0 | 0 |
| *Client 7* | 494 | 464 | 389 | 38 | 195 | 186 | 177 | 828 | 268 | 1332 |
| *Client 8* | 523 | 1827 | 71 | 150 | 481 | 756 | 403 | 332 | 323 | 23 |
| *Client 9* | 129 | 538 | 1211 | 1727 | 30 | 802 | 993 | 0 | 1 | 0 |

Table 2: Distribution of data samples (MNIST) from each class across 10 different clients in a federated learning setup.

| Class | 0 | 1 | 2 | 3 | 4 | 5 | 6 | 7 | 8 | 9 |
|-------|---|---|---|---|---|---|---|---|---|---|
| *Client 0* | 24 | 16 | 1150 | 73 | 77 | 247 | 558 | 798 | 867 | 953 |
| *Client 1* | 264 | 102 | 378 | 342 | 27 | 645 | 833 | 521 | 2020 | 411 |
| *Client 2* | 478 | 1000 | 758 | 964 | 418 | 371 | 215 | 150 | 744 | 511 |
| *Client 3* | 211 | 245 | 55 | 305 | 2746 | 389 | 363 | 28 | 346 | 2421 |
| *Client 4* | 1119 | 1893 | 1087 | 1167 | 183 | 30 | 127 | 641 | 0 | 0 |
| *Client 5* | 459 | 653 | 4 | 1418 | 70 | 1097 | 564 | 52 | 119 | 1140 |
| *Client 6* | 1023 | 497 | 126 | 257 | 859 | 204 | 2495 | 1963 | 0 | 0 |
| *Client 7* | 1147 | 1864 | 300 | 174 | 1450 | 440 | 81 | 2084 | 0 | 0 |
| *Client 8* | 959 | 89 | 1813 | 1026 | 9 | 1466 | 414 | 17 | 433 | 0 |
| *Client 9* | 239 | 383 | 287 | 405 | 3 | 532 | 268 | 11 | 1322 | 513 |

We constrain these dampening factors to not exceed the upper bound of $U$ to prevent model parameters from exploding in case of a user selecting $\lambda >> U$ values.

$$\beta = \min(\zeta, U) \tag{9}$$

The global model parameters are adjusted using this dampening factor as shown in equation 7, effectively removing the influence of the forget clients' data from the global model while preserving the updates of retained clients.

$$\theta'_i = \beta_i \cdot \theta_i \qquad \forall i \in |\theta| \tag{10}$$

where $\theta_i$ is the $i^{th}$ parameter of the global model, and $\beta_i$ is its corresponding dampening factor. The step-by-step workflow of the proposed method is outlined in Algorithm 1.

# 4 EXPERIMENTS AND RESULTS

## 4.1 EXPERIMENTAL SETUP

**Dataset and Baselines.** We assess the proposed method for client unlearning in a Federated Learning setup using various datasets, including MNIST LeCun (1998), CIFAR-10, and CIFAR-100 Krizhevsky et al. (2009). Our approach is compared against existing federated unlearning algorithms Fed-Eraser Liu et al. (2021) and PGA Halimi et al. (2022). The baselines also consist of a retrained model (by removing the *forget* client data) and a model trained with gradient ascent/negative gradient. The NegGrad is obtained by retraining the original model, where regular optimizers (such as Adam and SGD) are applied to the remaining clients, while gradient ascent is specifically used for the client whose data is to be forgotten. We use AllCNN+MNIST, Resnet18+CIFAR-10 and Resnet18+CIFAR-100 in our experiments. Our focus is on client unlearning, which is the primary form of unlearning required in non-IID Federated Learning setups.

**Evaluation Metrics.** We assess the effectiveness of a federated unlearning (FU) scheme by measuring its runtime and the accuracy with retrained model (by removing the forget client data) on both the retain (R-Set) and forget sets (U-Set). We further assess the unlearned models using two privacy attacks: backdoor attacks and membership inference attacks (MIA).

**Client-level Unlearning (non-IID FUL).** This paper addresses the challenge of client/sample-level unlearning, which is particularly difficult in a federated learning (FL) setup. Most existing research has focused on class-level unlearning, which is comparatively easier to manage. As a result, some of the performance outcomes and comparisons in our work may not seem as compelling, since sample-level unlearning is harder to validate. As noted in Wang et al. (2022),"*the sample-level unlearning task requires the model to remove specific data samples while maintaining the model's accuracy*". Despite these challenges, our results remain robust when this criterion is used for evaluation, demonstrating the effectiveness of our approach.

**Experiment Settings.** We created a set of 10 clients for each dataset. The data distribution inside each client for CIFAR-10, MNIST, and CIFAR-100 is given in Table 1, Table 2, Table 4, and Table 5, respectively. The hyper-parameters for FL are: the learning rate = 0.001, number of epochs = 100,

Table 3: Unlearning results in a federated learning setting. We use 10 client for CIFAR-10+ResNet18, MNIST+AllCNN and CIFAR-100+ResNet18. We forget *Client 0* in this experiment. The *cutoff ratio* in CONDA is set to 0.3 for CIFAR10 and 0.4 for MNIST and CIFAR-100 dataset. *accuracy & backdoor attack*: value closer to retrained model is *better*, *membership inference attack (MIA)*: value close to 50% or close to retrained model is *better*.

| Dataset | Metrics | Original Model | Retrained Model | NegGrad | PGA Halimi et al. (2022) | FedEraser Liu et al. (2021) | CONDA (OURS) |
|---|---|---|---|---|---|---|---|
| CIFAR-10 | Accuracy (R-Set) | $46.84 \pm 0.36$ | $44.96 \pm 0.03$ | $12.21 \pm 0.02$ | $40.56 \pm 1.20$ | $28.33 \pm 1.79$ | $\mathbf{41.44 \pm 1.55}$ |
| | Accuracy (U-Set) | $25.05 \pm 1.71$ | $20.59 \pm 0.75$ | $2.02 \pm 0.09$ | $15.64 \pm 1.60$ | $12.03 \pm 2.21$ | $\mathbf{21.86 \pm 2.03}$ |
| | Backdoor Attack | $35.41 \pm 0.55$ | $21.20 \pm 0.09$ | $0.00 \pm 0.00$ | $32.94 \pm 0.01$ | $18.82 \pm 0.03$ | $\mathbf{22.10 \pm 0.01}$ |
| | MIA | $94.85 \pm 0.03$ | $49.87 \pm 0.5$ | $49.96 \pm 0.05$ | $49.74 \pm 0.04$ | $50.09 \pm 0.04$ | $\mathbf{50.22 \pm 0.02}$ |
| MNIST | Accuracy (R-Set) | $97.85 \pm 0.00$ | $97.93 \pm 0.05$ | $83.54 \pm 0.35$ | $94.96 \pm 0.40$ | $46.43 \pm 0.00$ | $\mathbf{95.41 \pm 1.02}$ |
| | Accuracy (U-Set) | $83.99 \pm 0.01$ | $80.08 \pm 0.06$ | $57.19 \pm 0.25$ | $76.76 \pm 1.00$ | $39.98 \pm 0.47$ | $\mathbf{81.65 \pm 2.13}$ |
| | Backdoor Attack | $24.11 \pm 1.11$ | $0.21 \pm 0.21$ | $1.42 \pm 0.12$ | $1.21 \pm 0.06$ | $60.61 \pm 1.37$ | $22.16 \pm 0.02$ |
| | MIA | $95.18 \pm 0.01$ | $51.04 \pm 0.01$ | $50.17 \pm 0.02$ | $49.94 \pm 0.00$ | $48.36 \pm 0.00$ | $50.00 \pm 0.01$ |
| CIFAR-100 | Accuracy (R-Set) | $31.21 \pm 0.01$ | $31.52 \pm 0.04$ | $5.50 \pm 0.27$ | $30.10 \pm 0.13$ | $8.54 \pm 0.05$ | $28.67 \pm 1.31$ |
| | Accuracy (U-Set) | $29.54 \pm 1.00$ | $22.78 \pm 0.03$ | $0.56 \pm 0.00$ | $22.00 \pm 0.10$ | $8.73 \pm 0.00$ | $26.99 \pm 2.64$ |
| | MIA Accuracy | $94.82 \pm 0.01$ | $50.43 \pm 0.04$ | $49.88 \pm 0.00$ | $50.22 \pm 0.00$ | $50.00 \pm 0.00$ | $52.11 \pm 0.02$ |

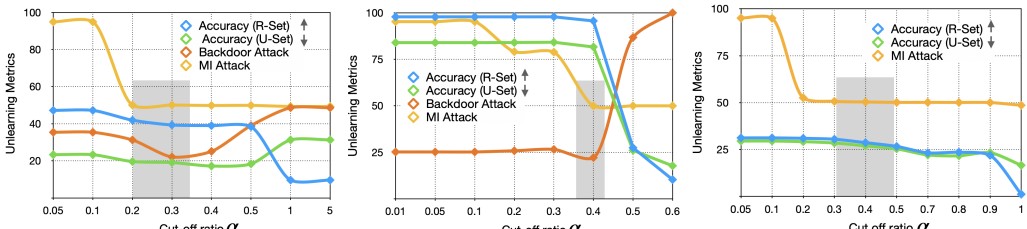

Figure 4: We show the results of CONDA on several Unlearning metrics: Accuracy (R-Set), Accuracy (U-Set), Backdoor Attack, and MI attack at different cut-off ratio $\alpha$. We visualize the unlearning plateau where R-Set accuracy, U-Set accuracy, Backdoor attack and MIA are near ideal values. Setting the $\alpha$ above or below the plateau leads to drop in desired unlearning performance. Results are shown in the order, CIFAR-10, MNIST, CIFAR-100 (left-to-right).

number of local epochs = 2. The remaining hyper-parameters follow state-of-the-art methods for fair comparison. For ConDa, the cut-off ($\alpha$) varies across datasets and distributions, the dampening constant ($\lambda$) is set to 10 for MNIST and 1 for CIFAR-10 and CIFAR-100, and the dampening upper bound ($U$) is 10 for MNIST and 1 for CIFAR-10 and CIFAR-100. We conduct an empirical analysis to examine the effect of varying the cut-off ratio $\alpha$ on the unlearning process across different datasets. We repeat each experiment three times and report the results along with the $\pm$ variance to account for fluctuations in performance.

**Challenges of Non-IID Federated Unlearning vs. IID Federated Unlearning.** A key challenge in federated learning is the varying data distribution among clients. In an IID distribution, where data from all classes is uniformly spread across clients, optimizers like SGD perform well, and unlearning is relatively straightforward, as it involves removing a small, evenly distributed subset of data. However, in real-world scenarios, assuming an IID distribution is unrealistic. In non-IID settings, certain classes may be concentrated within specific clients, leading to model biases. This makes unlearning more complex, as removing one client's data can disproportionately impact the model's learned features and overall accuracy. Additionally, interdependencies between clients' data further complicate isolating and unlearning specific contributions without adversely affecting others. While IID federated unlearning has been explored extensively, our experiments focus on tackling the complexities of non-IID federated unlearning.

## 4.2 RESULTS

The unlearning performance of CONDA is compared with existing methods in Table 3. A detailed discussion of the results and their significance is provided below.

**Accuracy.** The effectiveness of CONDA's unlearning is evaluated by comparing its accuracy on the forget client's data (U-Set) and the average accuracy for the retained clients (R-Set) against several baseline methods. Table 3 displays the results of our experiments across three different

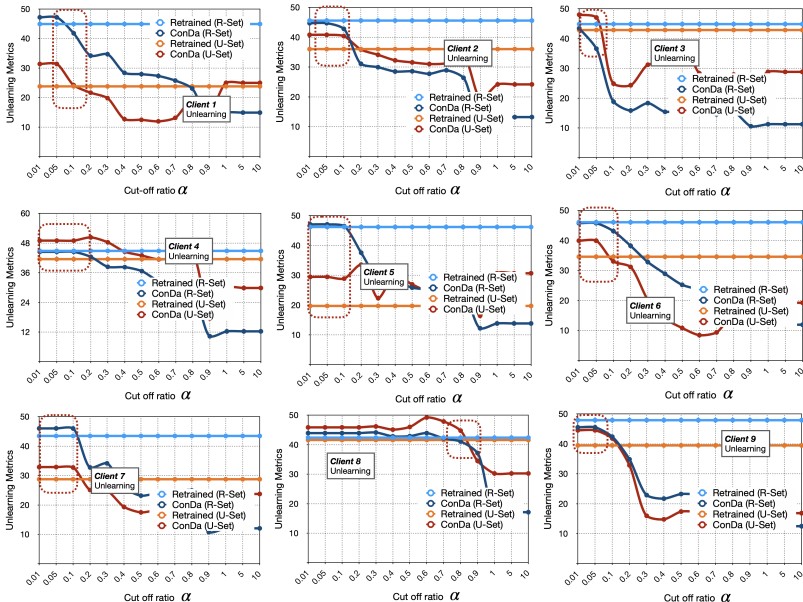

Figure 5: We present the results of CONDA for unlearning various clients (client 1 - client 9 in CIFAR-10) from the global model. These results are compared with the retrained model, which serves as the ground truth for unlearning. The performance of CONDA at different cut-off ratios $\alpha$ is displayed, with the optimal trade-off highlighted in the graph.

datasets where *Client 0* was designated for unlearning (U-Set). In CIFAR-10, the *retrained model* achieved an accuracy of 44.96 on the R-Set and 20.59% on the U-Set. CONDA obtains similar results with 41.44% accuracy on R-Set and 21.86% accuracy on U-Set. Compared to NegGrad, PGA, and FedEraser, our method achieves significantly better accuracy i.e., closer to the retrained model accuracy in both U-Set and R-Set. Similar trends are observed in both the MNIST and CIFAR-100 datasets. NegGrad, in particular, shows notably poor performance, highlighting its inability in handling unlearning within the challenging non-IID setting.

**Backdoor Attack.** To evaluate the privacy aspect of unlearning, we conduct backdoor attacks using the approach described in Gu et al. (2017). In this setup, backdoor triggers are introduced into part of the target client's dataset, making the global model vulnerable to these triggers and compromising its integrity. A successful unlearning method should reduce the model's accuracy on data with triggers while improving its accuracy on clean data. For all datasets, we introduce 500 backdoor samples, each containing a 40-pixel patch in the corner. The assigned labels are "1" for CIFAR-10 and "0" for MNIST.

The global model trained on a non-IID data distribution with backdoor triggers achieves a backdoor accuracy of 35.41% on CIFAR-10, correctly classifying 35.41% of test images with triggers (see Table 3). The goal of unlearning is to reduce this accuracy to match the retrained model's 21.20%—the gold standard. Among the evaluated methods, NegGrad fully neutralizes backdoor triggers with an accuracy of 0.0%, but it performs poorly on both R-Set and U-Set accuracy, making it impractical. Our CONDA achieves a backdoor accuracy of 22.10%, closely aligning with the retrained model, indicating effective backdoor mitigation. In contrast, PGA and FedEraser report backdoor accuracies of 32.94% and 18.82%, respectively. Similar results are observed on MNIST, confirming CONDA as an effective approach for mitigating backdoor vulnerabilities.

**Membership Inference Attack (MIA).** We employ Membership Inference Attacks (MIA) Shokri et al. (2017) as another evaluation metric, aiming to ensure that, after unlearning, an attacker cannot distinguish between examples that were unlearned and those that were never part of the training data, thereby safeguarding the privacy of the client requesting deletion. In an ideal defense, the attacker's accuracy would be 50%, signifying their inability to distinguish between the two sets, thus indicating the success of the unlearning method. Table 3 presents the MIA accuracy results across the three datasets. We note that all baseline methods achieve MIA accuracies close to 50%. Similarly, the

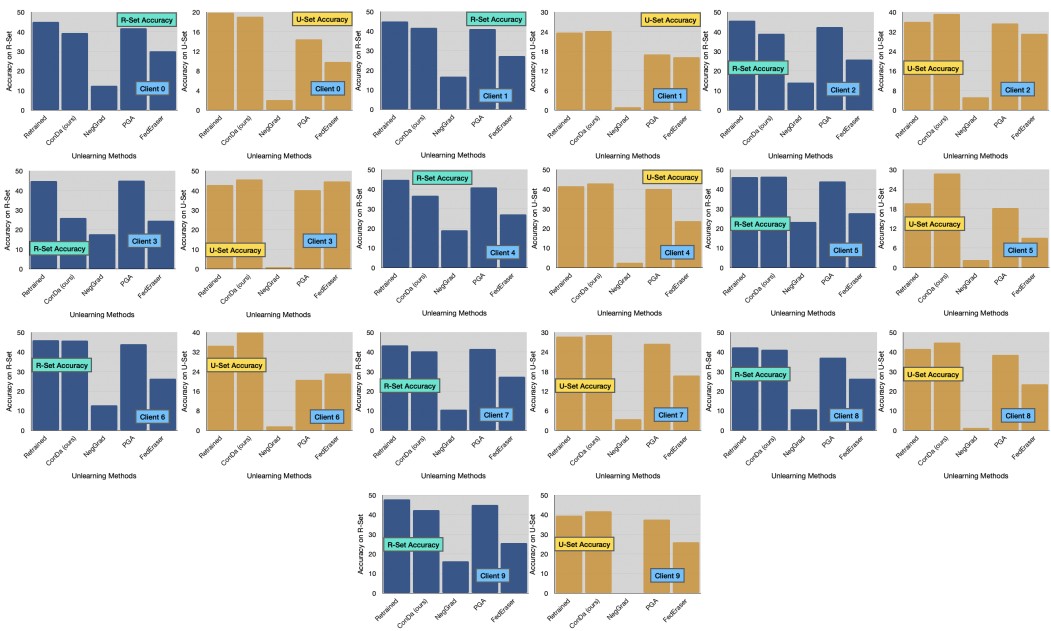

Figure 6: We compare the unlearning results of CONDA for various clients (client 0 - client 9 in CIFAR-10) and compare with the existing state-of-the-art methods. In most cases, CONDA closely follows the *Retrained* model in both U-Set and R-Set accuracy and performs much better than existing methods.

proposed CONDA reports MIA accuracies of 50.22%, 50.00%, and 52.11% for CIFAR-10, MNIST, and CIFAR-100, respectively.

**Runtime Comparison.** Runtime speed is crucial in federated machine unlearning because it directly impacts the system's ability to promptly remove sensitive or outdated information from the global model. Rapid unlearning minimizes the downtime of federated models, allowing them to quickly adapt to updated datasets while maintaining high performance. This is particularly important in dynamic environments, where data evolves continuously, and compliance with privacy regulations requires timely and effective data removal. Fast unlearning can also help in ensuring user privacy is protected in real-time, addressing data deletion requests efficiently.

In Figure 3, we compare the runtime of the proposed CONDA with the existing methods. We particularly focus on the time taken by different unlearning methods to unlearn the global model and observe that our method CONDA significantly outperforms other baseline methods. CONDA is approximately 5,882 × faster than PGA, approximately 3,584 × faster than FedEraser, and approximately 43,408 × faster than the NegGrad method. Our method is faster than all other methods. Moreover, unlike PGA and FedEraser, our method doesn't require data from the forget client and is free from any kind of fine-tuning. This significantly reduces the runtime of CONDA.

**Overview of Results and Take-Aways.** Our proposed CONDA demonstrates superior performance across all key evaluation metrics. It achieves accuracy results that closely match the retrained model on both the forget and retained sets, outperforming baseline methods while being robust to backdoor and MI attacks. Notably, it surpasses all methods in runtime efficiency, making it a highly practical and scalable solution for federated machine unlearning tasks. These results highlight CONDA's balance between privacy preservation, computational efficiency, and robust unlearning.

## 4.3 ABLATION ANALYSIS

**Impact of Cutoff-Ratio.** In equation 8, we introduced the cut-off ratio $\alpha$, which regulates the influence of the unlearning process on the model's updates, selectively dampening contributions from the forget clients. Figure 4 demonstrates the impact of varying $\alpha$ across different datasets. Our goal is to optimize performance by maximizing accuracy on the retained set (R-Set), minimizing accuracy on the unlearned set (U-Set), and ensuring that backdoor and MIA attack values are comparable to

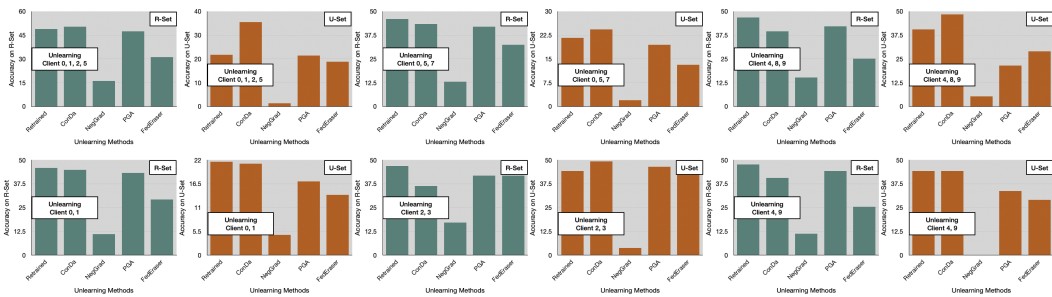

Figure 7: We show the results of multiple clients unlearning in CONDA and compare with the existing state-of-the-art methods. In most cases, CONDA closely follows the *Retrained* model in both U-Set and R-Set accuracy and performs much better than existing methods.

those of a fully retrained model. For CIFAR-10, the optimal cut-off ratio is $\alpha = 0.3$, effectively reducing backdoor effectiveness and bringing MIA accuracy close to the ideal 50%. For both MNIST and CIFAR-100, the best balance between R-Set and U-Set accuracy, along with optimal backdoor and MIA performance, is achieved at $\alpha = 0.4$.

**Unlearning Different Clients.** In Figure 4, we varied the cut-off ratio $\alpha$ for forgetting client 0 across all datasets. Figure 5 compares the R-Set and U-Set accuracy of the ConDa unlearned model with the retrained model for different clients on the CIFAR-10 dataset. $\alpha$ is varied from 0.01 to 10 to identify the optimal value under the non-IID data distribution shown in Table 1. The retrained model consistently serves as a benchmark, achieving maximum R-Set accuracy and minimum U-Set accuracy.

We find the optimal $\alpha$ for each client in CIFAR-10 and compare the results with the retrained model. For most clients, the optimal $\alpha$ falls between [0.01, 0.2], while for client 8, it ranges from [0.7, 0.9], indicating sensitivity to client-specific data distributions.

Next, we compare ConDa with the optimal $\alpha$ against baseline methods in Figure 6. ConDa achieves results closest to the retrained model, outperforming PGA and FedEraser, while NegGrad struggles to retain its R-Set accuracy

**Unlearning Multiple Clients.** In Figure 7, we apply ConDa to unlearn multiple clients on the CIFAR-10 dataset, comparing its R-Set and U-Set accuracy against baselines and the retrained model. Our results show that ConDa achieves a balanced trade-off between R-Set and U-Set accuracy when unlearning multiple clients. In contrast, methods like PGA and FedEraser tend to prioritize either U-Set accuracy or R-Set performance, often at the other's expense. NegGrad demonstrates lower accuracy in both R-Set and U-Set, highlighting the uneven effects of unlearning across different clients.

**Limitations.** Our method requires storing client contributions for all iterations on the server, leading to potential storage overhead, a common challenge in federated learning systems that track client contributions. Additionally, the cutoff ratio and dampening constant must be empirically selected for different datasets, introducing a practical limitation. However, our experiments show that these parameters typically lie within a predictable range, making the selection manageable. Future work could explore automated techniques for determining optimal parameter values, potentially using dataset-specific properties. While effective, further optimizations in memory management and parameter tuning could improve scalability and usability in larger real-world applications.

## 5 CONCLUSION

We introduced CONDA, a novel framework for fast and efficient federated unlearning through contribution dampening. Our method successfully removes client-specific information from federated models without retraining or requiring access to the remaining clients' data. Through extensive experimentation on multiple datasets, we demonstrated that CONDA achieves significant speedups compared to existing methods while maintaining robust model performance. By enabling the erasure of client data in federated learning systems, this work provides a vital tool for ensuring compliance with regulatory standards and addressing ethical concerns.

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

# A APPENDIX

## A.1 RELATED WORK

**Machine Unlearning.** Machine unlearning has gained considerable attention in recent years, driven by policies that grant users the right to erase their private data. Bourtoule et al. (2021) proposed SISA, a technique designed to improve unlearning efficiency. Golatkar et al. (2020) introduced effective unlearning strategies for deep neural networks. Tarun et al. (2023a) uses error maximizing noise generation and impair-repair weight manipulation techniques for unlearning. Tarun et al. (2023b) propose Blindspot unlearning method as a novel weight optimization process, useful for regression unlearning tasks. Chundawat et al. (2023a) proposed a teacher-student framework, where knowledge is selectively transferred between competent and incompetent teachers, resulting in a model that no longer retains information from the forget set. Chundawat et al. (2023b) demonstrated unlearning without relying on the original samples. Cotogni et al. (2023) introduces an unlearning method that leverages metric learning by guiding forget-set samples toward incorrect centroids in the feature space, with experimental evaluations demonstrating its effectiveness in both class-removal and homogeneous sample removal scenarios. Foster et al. (2024) proposes Selective Synaptic Dampening(SSD) method that identifies specific model parameters crucial for memorizing certain data, allowing for selective forgetting of targeted examples without compromising overall model performance. Kurmanji et al. (2024) introduces a scalable unlearning model that overcomes the limitations of prior methods by using a teacher-student framework to selectively forget data, while addressing scalability and performance issues in machine unlearning. Huang et al. (2024)proposes a meta-learning framework that balances forgetting and remembering in machine unlearning by leveraging feedback from a small subset of the remaining data and membership inference models to enhance generalization and optimize unlearning performance. More recently, Chatterjee et al. (2024) integrates continual learning and unlearning, using knowledge distillation to balance new information acquisition and selective forgetting. Sharma et al. (2024) proposes new evaluation metrics revealing limitations in current unlearning methods, advancing the understanding of concept erasure in diffusion models.

**Federated Unlearning.** Federated unlearning is an emerging field within machine unlearning, aimed at removing a specific client's data in federated learning systems due to privacy concerns, legal requirements, or the irrelevance of contributions. A basic method involves retraining the entire model from scratch, which is computationally expensive. Liu et al. (2021) introduced FedEraser, a recalibration method based on retained client contributions but requires client data during unlearning. Yuan et al. (2024) improves this by enabling the forgetting of multiple clients and dynamically releasing retained information, though it still involves client-side interaction. Halimi et al. (2022) employs Projected Gradient Ascent (PGA) for unlearning by maximizing the loss on forget data, while constraining model parameters within an L2 norm ball around the reference model, followed by fine-tuning to optimize performance. Su & Li (2023) optimizes retraining by dividing clients into clusters and only updating affected clusters, enhancing efficiency. Wu et al. (2022) reduced accuracy by directly subtracting forget client updates from the global model, but mitigated this with knowledge distillation using unlabeled data on global servers. Zhao et al. (2023) combines knowledge erasure

Table 4: Distribution of data samples (CIFAR-100) from each class across 10 different clients in a federated learning setup. Distribution of data from class 0 - class 49 for each client is shown below.

| Class | Client 0 | Client 1 | Client 2 | Client 3 | Client 4 | Client 5 | Client 6 | Client 7 | Client 8 | Client 9 |
|---|---|---|---|---|---|---|---|---|---|---|
| 0 | 118 | 22 | 29 | 130 | 31 | 36 | 2 | 63 | 64 | 5 |
| 1 | 135 | 47 | 28 | 182 | 7 | 2 | 68 | 8 | 13 | 10 |
| 2 | 4 | 7 | 40 | 24 | 5 | 5 | 209 | 140 | 5 | 61 |
| 3 | 111 | 53 | 128 | 49 | 7 | 17 | 3 | 4 | 98 | 30 |
| 4 | 15 | 42 | 144 | 0 | 117 | 72 | 2 | 1 | 84 | 23 |
| 5 | 1 | 17 | 9 | 1 | 23 | 163 | 101 | 159 | 6 | 20 |
| 6 | 130 | 2 | 44 | 3 | 123 | 1 | 28 | 13 | 127 | 29 |
| 7 | 0 | 32 | 69 | 8 | 40 | 4 | 92 | 210 | 12 | 33 |
| 8 | 2 | 119 | 169 | 45 | 2 | 2 | 119 | 19 | 20 | 3 |
| 9 | 2 | 102 | 106 | 4 | 14 | 51 | 80 | 84 | 16 | 41 |
| 10 | 100 | 49 | 83 | 16 | 130 | 80 | 0 | 2 | 17 | 23 |
| 11 | 5 | 206 | 61 | 18 | 2 | 81 | 32 | 10 | 76 | 9 |
| 12 | 9 | 17 | 25 | 214 | 18 | 20 | 118 | 67 | 2 | 10 |
| 13 | 80 | 22 | 0 | 13 | 1 | 129 | 3 | 55 | 93 | 104 |
| 14 | 23 | 164 | 57 | 76 | 73 | 45 | 36 | 23 | 1 | 2 |
| 15 | 0 | 2 | 2 | 103 | 39 | 21 | 69 | 76 | 13 | 175 |
| 16 | 57 | 19 | 5 | 1 | 88 | 213 | 50 | 40 | 2 | 25 |
| 17 | 155 | 102 | 2 | 6 | 24 | 2 | 23 | 80 | 76 | 30 |
| 18 | 82 | 40 | 5 | 1 | 26 | 56 | 60 | 6 | 141 | 83 |
| 19 | 1 | 7 | 6 | 7 | 180 | 15 | 1 | 46 | 183 | 54 |
| 20 | 49 | 9 | 62 | 1 | 65 | 91 | 14 | 143 | 62 | 4 |
| 21 | 128 | 49 | 84 | 19 | 67 | 6 | 5 | 3 | 95 | 44 |
| 22 | 45 | 61 | 29 | 33 | 8 | 121 | 57 | 133 | 6 | 7 |
| 23 | 105 | 4 | 89 | 4 | 27 | 18 | 25 | 121 | 41 | 66 |
| 24 | 53 | 41 | 1 | 7 | 65 | 15 | 123 | 42 | 32 | 121 |
| 25 | 54 | 54 | 2 | 15 | 25 | 31 | 3 | 20 | 233 | 63 |
| 26 | 8 | 57 | 23 | 38 | 28 | 1 | 104 | 85 | 126 | 30 |
| 27 | 48 | 17 | 22 | 183 | 4 | 47 | 15 | 29 | 134 | 1 |
| 28 | 18 | 39 | 8 | 15 | 139 | 91 | 86 | 67 | 3 | 34 |
| 29 | 0 | 39 | 93 | 4 | 70 | 3 | 205 | 5 | 38 | 43 |
| 30 | 30 | 33 | 47 | 17 | 72 | 77 | 19 | 76 | 60 | 69 |
| 31 | 0 | 33 | 71 | 20 | 96 | 158 | 61 | 38 | 22 | 1 |
| 32 | 52 | 17 | 119 | 18 | 7 | 2 | 11 | 30 | 79 | 165 |
| 33 | 5 | 3 | 24 | 77 | 0 | 36 | 230 | 86 | 22 | 17 |
| 34 | 44 | 34 | 72 | 35 | 3 | 15 | 213 | 50 | 6 | 28 |
| 35 | 110 | 5 | 114 | 103 | 11 | 51 | 44 | 23 | 12 | 27 |
| 36 | 6 | 113 | 1 | 19 | 66 | 32 | 1 | 6 | 82 | 174 |
| 37 | 27 | 12 | 15 | 19 | 50 | 2 | 19 | 111 | 28 | 217 |
| 38 | 5 | 77 | 21 | 6 | 86 | 29 | 54 | 8 | 42 | 172 |
| 39 | 128 | 145 | 95 | 3 | 26 | 3 | 10 | 33 | 24 | 33 |
| 40 | 1 | 26 | 167 | 119 | 14 | 23 | 8 | 63 | 28 | 51 |
| 41 | 1 | 44 | 5 | 15 | 72 | 0 | 89 | 44 | 146 | 84 |
| 42 | 14 | 14 | 2 | 132 | 16 | 61 | 2 | 28 | 23 | 138 |
| 43 | 55 | 7 | 12 | 122 | 5 | 122 | 4 | 2 | 80 | 49 |
| 44 | 36 | 1 | 50 | 7 | 72 | 97 | 49 | 23 | 38 | 127 |
| 45 | 0 | 61 | 48 | 56 | 85 | 123 | 4 | 19 | 55 | 47 |
| 46 | 10 | 57 | 85 | 13 | 105 | 12 | 95 | 47 | 109 | 8 |
| 47 | 47 | 2 | 122 | 38 | 76 | 103 | 12 | 47 | 5 | 148 |
| 48 | 17 | 16 | 16 | 4 | 1 | 24 | 3 | 10 | 9 | 4 |
| 49 | 99 | 42 | 0 | 0 | 53 | 57 | 145 | 18 | 61 | 21 |

and memory guidance to reduce discriminability for forgotten data while maintaining accuracy for retained clients. Li et al. (2023) introduces active forgetting by using randomly initiated teacher models to generate fake data, accelerating unlearning while preserving knowledge via Elastic Weight Consolidation (EWC). Fraboni et al. (2024) implements federated unlearning by leveraging an intermediate global model where client contributions surpass a predefined sensitivity threshold. It incorporates a novel Gaussian noise mechanism to perturb the intermediate model, ensuring effective and certified unlearning of the targeted clients.These methods demonstrate the evolving strategies for efficient unlearning in federated systems without compromising model performance. Other methods have been proposed for federated unlearning, each addressing different aspects of model optimization and efficiency (Che et al., 2023; Xiong et al., 2023; Zhang et al., 2023; Liu et al., 2022; Yuan et al., 2023).

Table 5: Distribution of data samples (CIFAR-100) from each class across 10 different clients in a federated learning setup. Distribution of data from class 50 - class 99 for each client is shown below.

| Class | Client 0 | Client 1 | Client 2 | Client 3 | Client 4 | Client 5 | Client 6 | Client 7 | Client 8 | Client 9 |
|---|---|---|---|---|---|---|---|---|---|---|
| 50 | 0 | 33 | 110 | 7 | 183 | 46 | 36 | 3 | 14 | 68 |
| 51 | 79 | 9 | 1 | 80 | 60 | 14 | 70 | 1 | 88 | 98 |
| 52 | 95 | 5 | 10 | 6 | 6 | 3 | 17 | 124 | 135 | 99 |
| 53 | 15 | 3 | 1 | 25 | 68 | 93 | 3 | 119 | 9 | 164 |
| 54 | 10 | 107 | 98 | 163 | 5 | 35 | 2 | 12 | 1 | 67 |
| 55 | 101 | 9 | 5 | 45 | 55 | 75 | 130 | 8 | 57 | 15 |
| 56 | 20 | 42 | 145 | 25 | 14 | 17 | 14 | 40 | 101 | 82 |
| 57 | 26 | 12 | 116 | 118 | 131 | 2 | 5 | 50 | 37 | 3 |
| 58 | 102 | 2 | 214 | 31 | 40 | 19 | 16 | 60 | 0 | 16 |
| 59 | 23 | 11 | 71 | 44 | 8 | 18 | 119 | 68 | 97 | 41 |
| 60 | 18 | 92 | 12 | 167 | 75 | 4 | 7 | 102 | 13 | 10 |
| 61 | 82 | 7 | 99 | 113 | 9 | 45 | 17 | 90 | 7 | 31 |
| 62 | 36 | 4 | 14 | 7 | 68 | 49 | 153 | 42 | 38 | 89 |
| 63 | 18 | 133 | 18 | 12 | 173 | 3 | 36 | 85 | 1 | 21 |
| 64 | 31 | 34 | 65 | 119 | 33 | 89 | 34 | 32 | 18 | 45 |
| 65 | 277 | 24 | 27 | 1 | 10 | 150 | 9 | 0 | 1 | 1 |
| 66 | 108 | 109 | 38 | 19 | 17 | 41 | 16 | 5 | 125 | 22 |
| 67 | 1 | 20 | 41 | 91 | 11 | 9 | 0 | 156 | 51 | 120 |
| 68 | 37 | 104 | 12 | 18 | 36 | 25 | 135 | 0 | 104 | 29 |
| 69 | 16 | 20 | 192 | 18 | 39 | 12 | 20 | 63 | 11 | 109 |
| 70 | 71 | 21 | 24 | 5 | 116 | 60 | 96 | 39 | 44 | 24 |
| 71 | 164 | 32 | 13 | 12 | 3 | 9 | 7 | 41 | 76 | 143 |
| 72 | 13 | 143 | 140 | 25 | 14 | 56 | 22 | 24 | 5 | 58 |
| 73 | 9 | 2 | 3 | 4 | 107 | 138 | 88 | 21 | 11 | 117 |
| 74 | 15 | 68 | 16 | 19 | 86 | 5 | 24 | 93 | 12 | 162 |
| 75 | 6 | 6 | 42 | 48 | 113 | 77 | 68 | 131 | 4 | 5 |
| 76 | 228 | 38 | 119 | 11 | 6 | 48 | 0 | 34 | 2 | 14 |
| 77 | 6 | 21 | 46 | 113 | 28 | 144 | 8 | 28 | 94 | 12 |
| 78 | 111 | 66 | 49 | 12 | 137 | 5 | 16 | 52 | 34 | 18 |
| 79 | 22 | 82 | 179 | 1 | 22 | 12 | 58 | 6 | 58 | 60 |
| 80 | 70 | 1 | 14 | 107 | 21 | 118 | 16 | 6 | 18 | 129 |
| 81 | 224 | 3 | 15 | 49 | 11 | 29 | 18 | 40 | 29 | 82 |
| 82 | 32 | 1 | 26 | 9 | 55 | 16 | 21 | 223 | 84 | 33 |
| 83 | 5 | 46 | 30 | 47 | 30 | 231 | 59 | 2 | 17 | 33 |
| 84 | 5 | 37 | 154 | 1 | 103 | 10 | 20 | 108 | 60 | 2 |
| 85 | 5 | 328 | 22 | 5 | 8 | 100 | 0 | 10 | 8 | 14 |
| 86 | 68 | 117 | 62 | 54 | 25 | 0 | 8 | 58 | 1 | 107 |
| 87 | 64 | 4 | 34 | 12 | 8 | 51 | 165 | 13 | 1 | 148 |
| 88 | 14 | 44 | 61 | 15 | 33 | 81 | 129 | 49 | 49 | 25 |
| 89 | 91 | 99 | 84 | 68 | 89 | 9 | 0 | 12 | 39 | 9 |
| 90 | 134 | 29 | 11 | 19 | 81 | 14 | 0 | 101 | 13 | 98 |
| 91 | 41 | 137 | 0 | 4 | 84 | 68 | 0 | 34 | 18 | 114 |
| 92 | 30 | 12 | 138 | 49 | 4 | 91 | 0 | 9 | 130 | 37 |
| 93 | 50 | 133 | 0 | 9 | 9 | 55 | 0 | 141 | 76 | 27 |
| 94 | 169 | 61 | 0 | 100 | 131 | 20 | 0 | 14 | 4 | 1 |
| 95 | 0 | 221 | 0 | 57 | 13 | 7 | 0 | 52 | 148 | 2 |
| 96 | 0 | 4 | 0 | 293 | 0 | 18 | 0 | 89 | 72 | 24 |
| 97 | 0 | 61 | 0 | 80 | 341 | 8 | 0 | 0 | 10 | 0 |
| 98 | 0 | 24 | 0 | 52 | 130 | 72 | 0 | 0 | 222 | 0 |
| 99 | 0 | 195 | 0 | 142 | 0 | 66 | 0 | 0 | 97 | 0 |

