# OpenReview forum: "ConDa: Fast Federated Unlearning with Contribution Dampening"
_ICLR.cc/2025/Conference — ICLR 2025 Conference Withdrawn Submission_

### Official Review · Reviewer_Jtx5 · 2024-10-31

**Soundness:** 2
**Presentation:** 2
**Contribution:** 2
**Rating:** 3
**Confidence:** 4

**Summary:**

The paper tackles the challenge of machine unlearning in a federated setup and aims to develop a better and more efficient unlearning method. According to the paper, the proposed approach avoids heavy computations (e.g., retraining the global model without the data of a forgetting client). Instead, it performs synaptic dampening, targeting the parameters influenced by the contributions of the forget client.

**Strengths:**

- Significant minimization of the computational overhead.
- The studied problem is relevant and important.

**Weaknesses:**

- The paper assumes the following:
   - *A non-IID FL setup is highly challenging to unlearn.*
   - However, non-IID FL is difficult in general federated optimization scenarios but not in federated unlearning.
   - Because of this claim, I suspect the method might not work in the IID setting since it creates another spectrum of problems associated with the similarity of local data distributions. Let's compare it to traditional machine unlearning, where you have class-level unlearning, which is similar to this setting of client-level forgetting in federated unlearning since you are assuming only a non-IID setting. Which, in turn, makes the problem solvable.

- Experimental settings:
   - You must consider more settings (data partitioning settings) by employing different kinds of non-IID-ness [1, 2]. There needs to be more than the settings provided in the paper (Tables 1, 2, 4, 5) to validate the method's validity.
   - Choice of the number of local epochs. Setting the local epochs to 2 in the non-IID setting favors overfitting each client's local data and makes them diverge more in the non-IID settings. This specific setting may help the proposed method obtain a better result since the gradients shared with the server from each client are diverse and easy to manage in the forgetting phase.

- The accuracy of CIFAR-10 (Table 3, Figures 4-7) is below 50%, which is unfavorable even though you use ResNet-18 architecture. It makes the validity of the results questionable.

- There are many hyperparameters ($\alpha, \lambda, U$), and no proper explanation or ablation study exists. The ablation study for the cut-off ratio ($\alpha$) is very sensitive, ranging from $0.01$ to $0.9$ [Line 511].

- [Line 204] Abuse of notation. Shouldn't the gradient update be multiplied by a learning rate scaler ($ - \dfrac{1}{\eta_{\ell}}$)?

- [Line 369] IID FUL has been explored extensively (you need to add a citation to back this up).

- The tables offer minimal insight into the non-IID-ness nature of the data partitioning. It would be more informative to describe the method used for generating the data splits and specify the random seed rather than including the tables, particularly those for CIFAR-100 (Tables 4 and 5). A heatmap could be a more effective alternative to the table for visualizing this information (only for a small number of classes).

- Table 3: The CIFAR-100 (Backdoor Attack) case is missing. According to the table, the proposed solution outperforms other methods in only half of the cases.

- The organization of the tables and figures can be improved. They can be better positioned with their first references in the paper.

[1] Li, Qinbin, et al. "Federated learning on non-iid data silos: An experimental study." 2022 IEEE 38th international conference on data engineering (ICDE). IEEE, 2022.

[2] Kairouz, Peter, et al. "Advances and open problems in federated learning." Foundations and trends® in machine learning 14.1–2 (2021): 1-210.

**Questions:**

See weaknesses.

---

### Official Review · Reviewer_bY6N · 2024-11-02

**Soundness:** 2
**Presentation:** 2
**Contribution:** 1
**Rating:** 3
**Confidence:** 4

**Summary:**

The goal of the paper is to unlearn the data that belonged to a given user/client that participated in the federated learning process. The proposed method dampens the parameters that were affected by the client whose data we aim at unlearning (it focuses on the client-level unlearning).

**Strengths:**

1. The paper presents a comprehensive evaluation of the proposed method.

**Weaknesses:**

1.	The main problem is that the design of the unlearning algorithm in this paper simply follows the work by Foster et al. 2024 [42] (Fast Machine Unlearning Without Retraining Through Selective Synaptic Dampening) and applies it to federated learning. There is the lack of description of how the method proposed in this paper differs from the Selective Synaptic Dampening (SSD) [42].
2.	In general this algorithm (compared to the original SSD) brings even more additional parameters and this becomes a non-trivial endeavor to find their optimal values. Algorithm 1 in [42] only has parameters $\alpha$ and $\lambda$, whereas this work adds the $U$ dampening upper bound.
3.	This method assumes that the server is fully trusted. However, in this case, the server can have access to the individual client’s data through, e.g., [1].
4.	The method assumes rather a big overhead on the server side to provide the unlearning. If the global model is large, then given millions of clients and hundreds of training rounds, the size of the collected data can be larger than the biggest LLMs.
5.	The results for the Retrained Model in Table 3 vs ConDa clearly indicate that the proposed method simply unlearns less (higher accuracy on U-Set – the forgotten set) and forgets more on the retained set (R-Set). ConDa also forgets much less about U-Set than other methods for unlearning, such as PGA and FedEraser. Further concerns regarding the evaluation are in the questions.

**References:**

1.	Boenisch, F., Dziedzic, A., Schuster, R., Shamsabadi, A. S., Shumailov, I., & Papernot, N. When the curious abandon honesty: Federated learning is not private. https://arxiv.org/abs/2112.02918

**Questions:**

1.	It is stated in the abstract: „ CONDA proves to be the fastest federated unlearning method, outperforming the nearest state-of-the-art approach by at least 100x” What is the meaning of the 100X?
2.	It is not clear what synaptic dampening means when reading the abstract. This is only known to the readers who know of the other paper "Fast Machine Unlearning Without Retraining Through Selective Synaptic Dampening" [42].
3.	How does the proposed FUL perform for the IID data?
4.	How is it ensured that the unlearning does not over-unlearn the data points so that they can still be detected as “previous” members using the membership inference attacks?
5.	Figure 1 is really unnecessary since it does not bring any value to the understanding of the paper.
6.	Tables 1 and 2 should be moved from the main paper to the appendix.
7.	On the other hand, the Related Work from the appendix should be moved to the main paper.
8.	Then, in the appendix, all the methods of federated unlearning should be described so that it is possible to discern what the novelty of the proposed method is, instead of stating at the end of A1: “Other methods have been proposed for federated unlearning, each addressing different aspects of model optimization and efficiency (Che et al., 2023; Xiong et al., 2023; Zhang et al., 2023; Liu et al., 2022; Yuan et al., 2023).”
9.	Table 3 reports results only for Client 0 being forgotten. What are the average results when each of the clients is forgotten? Figure 5 tried something like that but it is only per client and the results are not clear. These values should be averaged across all clients in Table 1.
10.	There is an inconsistency in how the method is referred to, for example, on page 7 it is either ConDa or CONDA (in italics). There should be a latex macro created for the name of the method and used consistently.
11.	Why do the results for the original paper on Selective Synaptic Dampening  [42] report better results, for example, for CIFAR100 in their Table 2 than in this paper in Table 1? For example, after the original SSD unlearning, the performance on the forget set was 0 whereas in this case, it was still 26.99. Moreover, the original SSD reported higher accuracy on the retain set.
12.	The places where Figures are presented vs where they are references are very different. For example, Figure 3 is shown on page 5 and referenced on page 9. There should be at most a +1/-1 page difference.
13.	In Figure 7, for clients 0, 1, 2, and 5, the performance of ConDa on U-Set is much higher than for the Retrained case. Why is this the case?

---

### Official Review · Reviewer_1qo9 · 2024-11-03

**Soundness:** 2
**Presentation:** 3
**Contribution:** 2
**Rating:** 5
**Confidence:** 3

**Summary:**

In this paper, the authors proposed a federated unlearning algorithm in which they tried to remove the effect of “opt-out” clients’ dataset on the global model via dampening the model weights those clients contributed the most. Their proposed algorithm needs additional storage to store past model updates sent by all clients across all global training rounds. They have conducted some numerical experiments to compare the performance of the proposed algorithm with several existing federated unlearning algorithms. They also performed some ablation studies to unveil how hyperparameter selection can affect the algorithm performance.

**Strengths:**

The paper studies an interesting problem on data unlearning for federated learning. The proposed algorithm is simple and easy to understand. The authors also compared the performance of the proposed algorithm with several baselines.

**Weaknesses:**

I listed a few weaknesses below:
1. It is not clear to me why the proposed algorithm is designed for Non-IID cases? What is the special component in the proposed algorithm that used to tackle Non-iid case? Why other existing unlearning algorithm cannot deal with non-iid case? Any intuition?
2. There is no analysis on the additional storage requirements. Does it scale linear with the number of clients, and model size, etc?
3. As the author pointed out in the limitation section, the proposed algorithm contains a few hyperparameters and they are all sensitive to clients’ datasets. In other words, hyperparameter tuning is needed every time to unlearn some client data from the global model, which makes the proposed algorithm not practical at all.
4. Storing all clients’ model updates across all global training rounds may leak clients’ private data information, especially in the non-iid setting. There is no discussion on this topic in the paper.
5. There is no theoretical analysis to provide any guarantee on the proposed algorithm that it does remove the effect of removed data samples, and to what extent it removes the effect of those points.

**Minor comment**: It is better to have the figure/table and the text discussing them at the same page or at most one page after. It is a bit hard to read through the text and scroll back to check the figures.

**Questions:**

Since we already use Eq. (9) to constrain the dampening factors what is the reason to use the cut-off ratio $\alpha$. It looks like they are all used for the same purpose: to control the dampening factors.

---

### Note · Authors · 2024-11-15

I have read and agree with the venue's withdrawal policy on behalf of myself and my co-authors.